# Why Oats Are Safe and Healthy for Celiac Disease Patients

**DOI:** 10.3390/medsci4040021

**Published:** 2016-11-26

**Authors:** Luud J. W. J. Gilissen, Ingrid M. van der Meer, Marinus J. M. Smulders

**Affiliations:** 1Wageningen University & Research, Bioscience, 6700 AA Wageningen, The Netherlands; luud.gilissen@wur.nl (L.J.W.J.G.); ingrid.vandermeer@wur.nl (I.M.v.d.M.); 2Wageningen University & Research, Plant Breeding, 6700 AA Wageningen, The Netherlands

**Keywords:** celiac disease, gluten-free, avenin, health claim

## Abstract

The water-insoluble storage proteins of cereals (prolamins) are called “gluten” in wheat, barley, and rye, and “avenins” in oat. Gluten can provoke celiac disease (CD) in genetically susceptible individuals (those with human leukocyte antigen (HLA)-DQ2 or HLA-DQ8 serotypes). Avenins are present at a lower concentration (10%–15% of total protein content) in oat as compared to gluten in wheat (80%–85%). The avenins in the genus Avena (cultivated oat as well as various wild species of which gene bank accessions were analyzed) are free of the known CD immunogenic epitopes from wheat, barley, and rye. T cells that recognize avenin-specific epitopes have been found very rarely in CD patients. CD patients that consume oats daily do not show significantly increased levels of intraepithelial lymphocyte (EIL) cells. The safety and the positive health effects of the long-term inclusion of oats in the gluten-free diet have been confirmed in long-term studies. Since 2009 (EC 41/2009) and 2013 (FDA) oat products may be sold as gluten-free in several countries provided a gluten contamination level below 20 ppm. Introduction of oats in the gluten-free diet of celiac patients is advised after the recovery of the intestine. Health effects of oat consumption are reflected in European Food Safety Authority (EFSA)- and Food and Drug Administration (FDA)-approved health claims. Oats can form a healthy, nutritious, fiber-rich, and safe complement to the gluten-free diet.

## 1. Introduction

Oats have a long history of use as a nutritious substance in food and feed. Excavations from 32,600 years ago in Italy revealed thermal pre-treatment and grinding of, specifically, oat seeds by humans [1], indicating that cereal food technology may have been developed long before that date. However, the cultivation of oats as a crop developed much later than that of wheat. It has been suggested that the common oat (*Avena sativa*) spread as a weed impurity of wheat and barley seeds between four and five thousand years ago, in the Bronze Age, from the Near East to Central and North Europe. Once arrived, oat turned out to adapt well to the cool and humid climate and long day length (Figure 1), and the “weedy” oat became domesticated in Europe by the early farmers into heterogeneous, robust landraces that were passed on from generation to generation; some of these landraces have survived up to modern times [2]. 

Oat also presented itself as a perfect feed for working horses in agriculture and therefore became often on-farm cultivated, year after year. Oat behaved as an “… attractive and productive servant, thriving on neglect and disinterest …”, but was gradually overshadowed by more assertive but less attractive step-crops (e.g., wheat) [3,4]. As a consequence, oat breeding lags behind that of wheat and barley, and attention to the pureness of the sowing seed is still limited. Contamination with wheat and barley is therefore high. For example, the U.S. specification for No. 1 oats allows the presence of up to 2% foreign material, which could be all wheat and barley [5]. Oat is undervalued, despite its unique composition, with many nutrients that contribute to health and reduce the risk of the incidence of degenerative diseases. Furthermore, oat fits into a healthy gluten-free diet. These issues will be elaborated here.

## 2. The Journey of Oats towards Its Gluten-Free Status

During the shortage of bread in the final year of World War II, a significant health improvement was observed in The Netherlands in some children suffering from CD. These young patients quickly deteriorated when planes from the Allied Forces dropped bread loafs. Based on this observation the exclusive role of gluten was discovered and documented a few years later by Dicke [6,7]. Initially, CD-causing gluten was mentioned to originate from wheat, barley, rye, and oat. In the 1990s several feeding studies have shown that patients with CD in general tolerate oats without signs of intestinal inflammation. However, there was an exception. In an open challenge study of adult CD patients using pure oats, three patients developed villous atrophy [8]. They had developed mucosal T cells that showed avenin-reactivity in vitro; such cells are responsible for mucosal inflammation in situ. It was therefore suggested that oat intolerance may be a reason for villous atrophy and inflammation in CD patients who are eating oats but otherwise adhere to a gluten-free diet [9]. This study had a long echo and created doubt not only with the individual patients and their patient associations, but also in official organizations such as the European Commission and the Codex Alimentarius that initially put “oats and products thereof” on the allergen list, next to wheat, barley, and rye. Remarkably, the Norwegian study that described the three patients reacting to oats has not been confirmed by other studies on CD patients, although it is likely that more patients may exist that react to oats (Lundin, personal communication).

Gradually, it became clear that the most serious problem for CD patients in consuming oat products was the frequent contamination with gluten-containing cereal material that may occur in each step of the production chain, involving sowing seed purity [4], cultivation and harvesting practices, milling, and further processing into food products [10,11,12,13,14,15,16,17]. There is now sound scientific evidence that CD patients can regularly eat up to 100 g/day of uncontaminated oats without any harm. This is based on long-term studies on cohorts of CD patients regularly eating oats, e.g., in Scandinavia [18,19,20,21]. In 2016, a Canadian position paper also concluded that oats uncontaminated by wheat, barley, and rye can be safely ingested by most patients with CD and that there is no conclusive evidence that the consumption of uncontaminated or specially produced oats containing no greater than 20 ppm gluten by patients with CD should be limited to a specific daily amount. They advise introducing uncontaminated oats in the gluten-free diet after all symptoms of CD have resolved [22], because not yet fully recovered intestines may be especially sensitive to the high fiber content of oats. Note that gluten-free diets are generally very low in fiber [20]. It is fair to say that a large-scale and international intervention study (although not intended as such) regarding the safety of commercially pure oat products, produced from an unknown number and diversity of oat varieties, has been ongoing for at least ten years in the form of a steadily increasing daily pragmatic practice of oat consumption by CD patients in many countries [23]. 

The scientific data on the very low appearance of intolerance to oat, i.e., the numerous publications that show the safety of oats to the general CD patient population, have been governmentally recognized. In January 2009, the European Commission Regulation (EC) 41/2009 on the content and labeling of foods for individuals with CD came into force in Europe. Oat products containing less than 20 ppm gluten are now allowed to be sold as gluten-free and may carry the official logo of the Association of European Coeliac Societies (AOECS) on a contract basis and regular audit of the producer. Since August 2013 also the U.S. (www.federalregister.gov) allows oats to be sold as gluten-free, provided any contamination with gluten from wheat, barley, and rye is below 20 ppm, thus in line with the Commission Regulation (EC) 41/2009. According to their Food Standard Code, Australia and New Zealand are still reluctant, but this may change rapidly soon since a consortium of Australian researchers has now concluded that the low rates of T cell activation after a substantial oat challenge (100 g/day) in a representative number of patients suggest that doses of oats commonly consumed are insufficient to cause any clinical relapse [20].

## 3. Recent Safety Objections Refuted

Traditionally, the term “gluten” applied only to wheat; however, because the rye and barley storage proteins have similar toxic sequences, the celiac disease community (patients and physicians) adopted the term “gluten” for all proteins active in celiac disease, i.e., the prolamins from wheat, barley, and rye [24,25,26]. Several factors explain why oats are basically safe for CD patients: (1) The content of avenin protein in oat grains is small (10%–15% of total protein) compared to the level of gluten in wheat, barley, and rye. This coincides with a low number of genes (maximum ten genes in the hexaploid species *Avena sativa* compared to at least 100 gluten genes in wheat, most of which are gliadin genes) [24,25,26,27]. (2) None of the currently known epitopes from wheat, barley, and rye occur in oats ([23], Table 1). The two avenin-specific epitopes [9] exist in all oat varieties and species [23], but very few CD patients react to these, making oat intolerance a rare event [8]. (3) Variants of the immunogenic epitopes of wheat, barley, and rye occur in oats, but they differ at two or three amino acid residues, which will abolish T cell binding completely or nearly completely, depending on the position of the substitutions in the nine-amino acid T cell epitope [28]. Avenin cross-reactivity by T cells induced following an oral wheat challenge in vivo was not found [20]. As these oat peptides are also sensitive to digestion by pepsin, trypsin, and chymotrypsin in the gastrointestinal tract, they are unlikely to have any clinical relevance.

Nevertheless, there seems to be still an “oat dispute” going on, in which it is suggested that different oat varieties may cause different immunological responses in people suffering from CD. These differences were concluded on the basis of assays with the gluten-specific R5 and G12 monoclonal antibody (mAb) tests, and on the basis of in vitro T cell proliferation [29,30,31]. Antibody tests are part of the standard protocol for the detection of gluten according to the Codex Alimentarius (R5 antibody-based tests) or opting to become accepted (G12). Both antibodies have been developed for the detection of wheat gliadins, and a multiplication factor should be applied in the protocols to estimate the actual gluten content. Arithmetical adaptations are also required when these tests are applied to quantify the gluten content in barley and rye products. 

Both tests recognize short sequences of five (R5) or six (G12) amino acid residues from wheat gliadins [23]. Complete CD immunogenic epitopes are always sequences of nine amino acid residues; these sequences have been accurately described and named [32], and the binding interactions between antigen-presenting cells (APC), the gluten peptide, and the T cell receptor have been modeled and are well understood [33]. Hence, the mAbs do not directly measure epitopes. At best, these tests can be an indirect indication of immunogenicity in cases where the sequence is part of a known epitope and the frequency and expression of that epitope sequence has been determined [34,35]. This situation is clear for the R5 antibody with regard to the 26-mer and 33-mer of wheat gliadins, and it may also have relevance for sequences of closely related genera such as barley and rye because of the high homology of the prolamins. The situation is different for oat avenins. In the entire genus *Avena*, no perfect recognition sites of the R5 and the G12 mAbs are present, nor are any of the CD epitopes from wheat, barley, and rye (Table 2) [23]. Some of the sequences that may be recognized by G12 could be linked to the two described avenin epitopes [23], but these most likely occur in every oat variety. Thus, the immunogenic responses of the G12 mAb to oat extracts of some varieties and less to other varieties [29,30,31] are most likely the result of cross-reactivity with some sequences in avenins, and must be considered to be irrelevant with regard to the presence/absence of the identified clinically toxic sequence profile of the list of internationally agreed epitopes [25]. Gluten detection using the G12 antibody therefore gives misleading results in oats.

The cross-reactivity of the G12 mAb to oats also complicates the detection of gluten contamination in oats, while the R5 has no cross-reactivity and gives a clear-cut outcome. Compared to both, a new antibody test has recently been put on the market that perfectly recognizes one important gliadin epitope of wheat (the alpha-gliadin glia-α20 epitope, or DQ2.5-glia-α3) and that does not cross-react with oat (avenins) sequences (Gluten-Tec Elisa, Europroxima, Arnhem, The Netherlands). The test includes a synthetic peptide that encompasses the α20-gliadin epitope as an internal standard for correct quantification. An alternative way of detecting epitope-containing peptides in a laboratory setting is using liquid chromatography-multiple reaction monitoring mass spectrometry (LC-MRM/MS), as has been developed for wheat gliadin epitopes [36].

Similarly, caution should be exerted when extrapolating the results of in vitro T cell assays. Indeed, CD patients may possess T cells capable of responding to immuno-dominant avenin peptides ex vivo, but the frequency and consistency of these T cells in blood is very low, as is their activation rate after consumption of doses of oats commonly consumed (even up to 100 g/day) [20]. Hence, the relation between antibody signals and T cell responses as established for gliadins in wheat and prolamins in wheat, barley, and rye should not simply be extrapolated to oat. Without confirmation by other test methods, and from feeding challenges and in situ measurements, data on presumed differences in toxicity of oat varieties are misleading to the overall patient population. They may jeopardize the trust of patients in oat consumption, and they could lead to screening and selection or rejection of oat varieties without a proper scientific basis. 

## 4. The Contribution of Oats to a Healthy Gluten-Free Diet

The health advantages of the consumption of oats are generally recognized. Research on the health benefits of oats has been accumulated over the last three decades. Especially, the hypocholesterolemic properties of oats have been documented. Other data also describe the cardiovascular benefits of oats beyond lowering cholesterol levels through positive effects on the blood glucose level, and through better management of body weight and blood pressure. These characteristics are mainly attributed to the high content of oat-specific beta-glucans, which are soluble food fibers. In addition, oat fibers increase the fecal bulk, which contributes to a normal stool, and have a positive impact on the functioning of the microbiome. Several health claims on these topics have been approved by the Food and Drug Administration (FDA) in the U.S. and the European Food Safety Authority (EFSA) in Europe (see Box 1). These are described and explained in detail elsewhere [37]. Oats are generally consumed as whole grain products. Currently, the interest in whole grain foods is rapidly increasing. The intake of whole grain and cereal fiber has been inversely related with the risk of chronic diseases and with reduced total and cause-specific mortality [38,39].

Box 1.Approved European Food Safety Authority (EFSA) health claims relevant to oats (see, e.g., ref [36]).Beta-glucans (3 g/day) contribute to the maintenance of normal blood cholesterol levels (EU 432/2012)Consumption of beta-glucans from oats and barley as part of a meal (4 g/30 g carbohydrates) contributes to the reduction of the blood glucose rise after that meal (EU 432/2012)Oat grain fiber contributes to an increase in fecal bulk (EU 432/2012)Reducing consumption of saturated fat contributes to the maintenance of normal blood cholesterol levels (EU 432/2012)Oat beta-glucan (3 g/day) has been shown to actively lower/reduce blood cholesterol. High cholesterol is a risk factor in the development of coronary heart disease (EU 1160/2011)

Lesser-known oat-specific bioactive compounds are the avenanthramides (phenolic amine conjugates) [40], which give oats their anti-inflammatory properties through suppression of prostaglandine E2 [41], have a strong antioxidant capacity, and show antihistamine activity. These compounds also suppress the proliferation of vascular smooth muscle cells, a process known to be a contributing factor in the development of atherosclerosis [42].

The protein content in the oat groat is relatively high (15%–20% by weight). The digestibility of oat protein is high (90%), which is comparable to the protein digestibility of rice and corn. The majority (85%–90%) of oat protein consists of globulins, in contrast to wheat in which the vast majority consists of gluten. The composition of oat proteins fits well to the human needs of essential amino acids; the amounts of lysine and threonine in whole grain oats almost meet the dietary requirements (80% for both amino acids [43]).

Starch is the major component of the oat grain: 60% of the total dry weight. The amylopectin:amylose ratio is about 3. The digestibility of oat starch is about 100%. Oat starch digests slowly, partly due to the presence of high amounts of fiber and the high oil content in whole grain oat, which retards stomach emptying and improves digestion. This results in a gradual supply of glucose to the intestine, which maintains a long feeling of satiety. As a result, whole grain oat foods have a low glycemic index (GI), which is advantageous in cases of diabetes and obesity [37,44].

Compared to others cereals, oat grains have a relatively high oil content of on average 7% oil, but some varieties can have up to 18% [45]. The three most abundant fatty acids are palmitic (C16:0; 20%), oleic (C18:1; 35%), and linoleic (C18:2; 35%) acids, which account for about 90%–95% of the total fatty acids. The largest part is thus unsaturated, but there is more omega-6 than omega-3. Alpha-linolenic acid (18:3, omega-3) is notably present in the oat germ. The high lipid content can also have an adverse effect on the sensory quality of oat products because of lipid oxidation, which produces fatty acid hydroperoxides and volatile aldehydes, causing rancidity. Therefore, before further processing, oats are kilned—a high-temperature treatment to inactivate especially lipases.

The awareness that whole grain oat products have the potential to positively impact many health-related conditions associated with coronary heart disease, diabetes, satiety/weight management (low glyceamic index [GI]), and blood pressure is still limited among consumers as well as in the medical field. Substantiated health claims may be helpful to educate the consumer on the important relationship between healthy eating behavior and disease prevention. However, these health claims have to be brought to public attention through targeted communication, advertisement, and product labeling. 

## 5. Conclusions

Long-term cohort studies as well as short-term intervention studies have revealed the safe consumption of oats by celiac disease patients, provided that the products are uncontaminated with wheat, barley, or rye, or are specially produced to avoid gluten contamination above 20 ppm (the international legally agreed threshold). None of the currently known epitopes from wheat, barley, and rye are present in oats. Two avenin-specific epitopes, which can be present in all oat varieties and species, have been found to be reactive in only a very few patients. Immunological responses of some oat varieties in the gliadin-specific monoclonal antibody are considered as a result of cross-reactivity and differences in antibody signals among oat varieties and have no clinical relevance as the T cells do not show cross-reactivity to avenins. Gluten-specific antibody signals should therefore not be extrapolated to oat avenins. It is opportune to focus on the numerous health benefits of whole grain oats, which contribute significantly to the nutritional quality of the gluten free diet for CD patients in which the intestine has recovered. In general, the agriculture and food industries, hand in hand with the medical and healthcare sector, should seize this opportunity to reduce the incidence of chronic diseases and to increase the robustness of the entire population, including the people with CD.

## Figures and Tables

**Figure 1 medsci-04-00021-f001:**
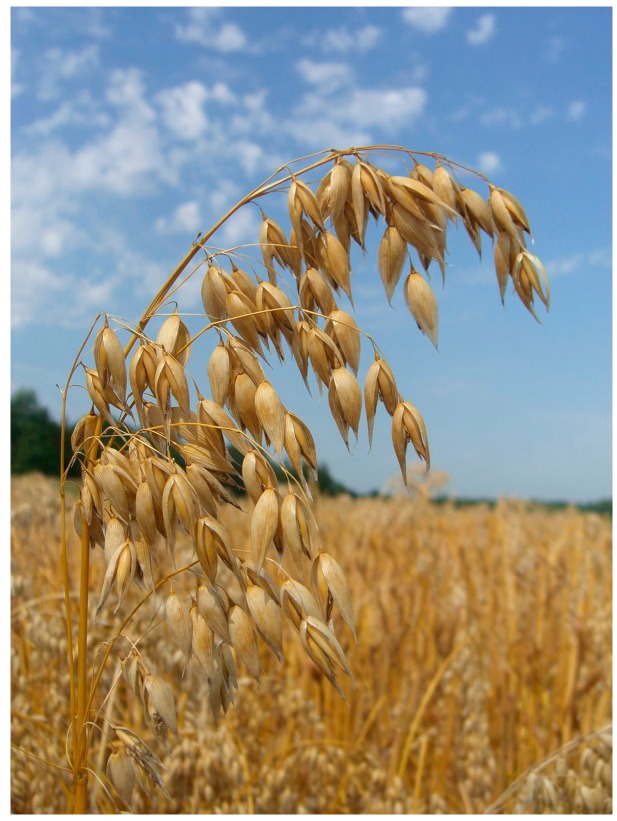
Oat panicle in an agricultural field.

**Table 1 medsci-04-00021-t001:** Variants of T cell epitopes present in oat avenins that are predicted to resist trypsin and chymotrypsin digestion.

T Cell Epitopes (Deamidated in Bold)	Epitope Variants (Amino Acid Differences Underlined) Present in Oat Avenins that Are Predicted to Resist Trypsin and Chymotrypsin Digestion
DQ2.5-glia-γ2 IQPEQPAQL (from wheat)	N Q P Q Q Q A Q F
	I Q P Q Q L P Q Y
DQ2.5-glutL2 FSQQQESPF (from wheat)	V *Q* Q Q Q Q Q P F
DQ2.5-hor-1 PFPQPEQPF (from barley)	P Y P E Q Q Q P F
DQ2.5-sec-1 PFPQPEQPF (from rye)	P Y P E Q Q Q P F
DQ2.5-ave-1a PYPEQEEPF (from oat)	P Y P E Q Q Q P I
DQ2.5-ave-1b PYPEQEQPF (from oat)	P Y P E Q Q Q S I
	P Y P E Q Q Q Q L

Perfect T cell epitopes from wheat, barley, and rye are not present in oat avenins, but variants with one, two, and three amino acid differences (underlined) can be found in avenin sequences. Of 89 proteins derived from genomic sequencing of 13 diploid, tetraploid and hexaploid *Avena* species, only these eight epitope variants resisted in silico trypsin and chymotrypsin proteolysis; none of these variants remained intact if pepsin was also added (adapted from Londono et al. [23]).

**Table 2 medsci-04-00021-t002:** Variants of the recognition sites of the R5 and G12 antibodies with one and two amino acid substitutions in oat avenins. No perfect recognition sites were present in avenin gene sequences of 13 *Avena* species (adapted from Londono et al. [23]).

Antibody	Reported Recognition Site in Wheat	Most Similar Variants that Exists in Oat (Substitutions Underlined)	Occurrence (% of Sequences)
R5	QQPFP	Q Q P F L	23.4
	QQPFP	Q Q P F V	27.6
	QQPFP	Q Q P F M	23.4
	QQPYP	Y Q O Y P	100
G12	QPQLPY	Q P Q L Q Q	73.4
	QPQQPY	Q P Q Q Q A	40.4
	QPQQPY	Q Q Q Q P F	48.9
	QPQQPY	Q P Q Q L P	14.9
	QPQQPY	Q P Q Q L S	6.4
	QPQLPF	Q P Q L Q L	8.5

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
