# Peer review of "Why Oats Are Safe and Healthy for Celiac Disease Patients"

_medsci, 2016, doi:10.3390/medsci4040021_

Reviewer 1 Report

I l;ike this review. It covers some topics that are slightly controversial and dissects them impressively, providing a definitive analysis.  It is well written. I think it is a valuable contribution to the literature on oats and oat proteins in relation to celiac disease.

I have only a few minor comments for the authors' consideration.

Line 38. I would prefer "may have been developed" to "must have been developed."

Lines 79-82. I think this sentence could be re-written for clarity. To me, it seems to say that contamination of oats by wheat gluten and  and gluten-containing cereal material occurs only during post-harvest processing, but this contamination also can occur during the growing of oats.

lines 111,112. It might be helplful at this point to explain that traditionally the term 'gluten' applied only to wheat, but because the rye and barley storage proteins have similar toxic sequences to those of wheat, even though a gluten ball cannot be washed from rye and barley, the celiac disease community (patients and physicians) adopted the term 'gluten' for all the proteins active in celiac disease. See: B. Jabri et al. Immunological Reviews 2005, 206:219-231 (page 222)

Line 113. I think the number 50-100 gliadin genes would be more appropriate for gluten rather than gliadin (see F.M.Dupont et al. Proteome Science 2011, 9:1)

Author Response

Line 38. I would prefer "may have been developed" to "must have been developed."

> This has been changed.

Lines 79-82. I think this sentence could be re-written for clarity. To me, it seems to say that contamination of oats by wheat gluten and  and gluten-containing cereal material occurs only during post-harvest processing, but this contamination also can occur during the growing of oats.

> the sentence has been rewritten.

lines 111,112. It might be helplful at this point to explain that traditionally the term 'gluten' applied only to wheat, but because the rye and barley storage proteins have similar toxic sequences to those of wheat, even though a gluten ball cannot be washed from rye and barley, the celiac disease community (patients and physicians) adopted the term 'gluten' for all the proteins active in celiac disease. See: B. Jabri et al. Immunological Reviews 2005, 206:219-231 (page 222)

> We have added a sentence explaining this, with this reference and another one.

Line 113. I think the number 50-100 gliadin genes would be more appropriate for gluten rather than gliadin (see F.M.Dupont et al. Proteome Science 2011, 9:1)

> This has been reformulated.

Reviewer 2 Report

The aim of the manuscript was to provide an evidenced-based review of the current literature pertaining to the safety of oats in patients with celiac disease. The authors minimize the impact of an immune response following cross-reactivity while it is well described the potential pathogenetic relevance of Molecular mimicry. Such triggering role and its pathogenetic importance have been investigated and demonstrated for many autoimmune diseases. It is important that the authors discuss this point in their conclusion or provide more arguments on the no clinical relevance of the cross-reactivity of immunoligical response between oat and wheat in some patients.

The paragraph on page 5 line 168 starting ‘’Here, the scientific (medical) world… should be deleted because it is out of the context of the review.

Author Response

The aim of the manuscript was to provide an evidenced-based review of the current literature pertaining to the safety of oats in patients with celiac disease. The authors minimize the impact of an immune response following cross-reactivity while it is well described the potential pathogenetic relevance of Molecular mimicry. Such triggering role and its pathogenetic importance have been investigated and demonstrated for many autoimmune diseases. It is important that the authors discuss this point in their conclusion or provide more arguments on the no clinical relevance of the cross-reactivity of immunoligical response between oat and wheat in some patients.

> We do not use the term Molecular mimicry. We revised the text so as to be clear that the avenins do not elicit a response by T cells in vivo, so they do not give an immunological response. The in vitro reactions that occur are therefore considered as cross-reactivity.

The paragraph on page 5 line 168 starting ‘’Here, the scientific (medical) world… should be deleted because it is out of the context of the review.

> We have deleted the paragraph.

Reviewer 3 Report

This short review article provides an update on the current status of oats consumption in celiac disease. They state that due to the health benefits provided by oats, and the number of long-term studies suggesting safety, that most CD patients can safely eat oats. Importantly, it highlights that caution must be taken with regards to previous interpretations that oats varieties exhibit distinct levels of immunogenicity based on antibody binding. Use of antibodies for detection of oats in food is now mentioned as a feature of commercial kits utilizing the G12 Ab - is this misleading?  

The authors do not provide a clear opinion for the process of introduction of oats to the gluten free diet. Do they believe oats introduction should still be managed clinically with the help of health providers? Or do they believe CD patients can immediately introduce oats to the diet upon diagnosis? Such views would be useful in this review article.

References are up-to-date and appropriate

Author Response

This short review article provides an update on the current status of oats consumption in celiac disease. They state that due to the health benefits provided by oats, and the number of long-term studies suggesting safety, that most CD patients can safely eat oats. Importantly, it highlights that caution must be taken with regards to previous interpretations that oats varieties exhibit distinct levels of immunogenicity based on antibody binding. Use of antibodies for detection of oats in food is now mentioned as a feature of commercial kits utilizing the G12 Ab - is this misleading?

> Yes, we present arguments why the G12 Ab results can be considered misleading. We have now added a short sentence that includes the term ‘misleading’ for clarity.

The authors do not provide a clear opinion for the process of introduction of oats to the gluten free diet. Do they believe oats introduction should still be managed clinically with the help of health providers? Or do they believe CD patients can immediately introduce oats to the diet upon diagnosis? Such views would be useful in this review article.

> We added the sentence that it is advised to introduce oats after recovery of the intestine. Often such patients will still be under medical care.